# Fecundity, Length at First Sexual Maturity and Gonadal Development of *Lepidorhombus boscii* in the Eastern Adriatic Sea

**DOI:** 10.3390/biology12010131

**Published:** 2023-01-13

**Authors:** Nika Ugrin, Antonela Paladin, Svjetlana Krstulović Šifner

**Affiliations:** 1Department of Marine Studies, University of Split, 21000 Split, Croatia; 2Faculty of Science, University of Split, 21000 Split, Croatia

**Keywords:** *Lepidorhombus boscii*, reproductive biology, maturation, histology, gonadosomatic index, Adriatic Sea

## Abstract

**Simple Summary:**

*Lepidorhombus boscii* is a flatfish of high commercial value. However, the existing knowledge about the biology and ecology of this species is scarce. The aim of this study is to describe the reproductive cycle of *L. boscii* in the eastern Adriatic Sea. The sex ratio of *L. boscii* was 1:0.89 (female:male), and the size at first sexual maturity was 19.2 cm for females and 20.3 cm for males. Previtellogenic and vitellogenic oocytes are present in the ovaries throughout the year. *L. boscii* has a prolonged spawning season in the Adriatic Sea with the main spawning season between November and March.

**Abstract:**

This study presents data on the seasonal changes in population structure, maturation, fecundity and reproduction of the *Lepidorhombus boscii* from the central-eastern Adriatic Sea. This species is a commercially valuable fish in the Adriatic Sea; still, knowledge about its reproductive biology is very scarce. The analyses are based on the data collected between July 2020 and June 2021 by bottom trawls. In total, 963 individuals were collected out of which 508 were females (47.24%) and 455 were males (52.75%). Throughout the year, oocytes in the previtellogenic and vitellogenic stages were present in the ovaries, and the development of the ovaries followed the values of the gonadosomatic index. *L. boscii* has asynchronous ovarian development and a prolonged spawning season in the Adriatic Sea with the main spawning season between November and March. The oocyte diameter ranged from 34.01 to 562.72 μm (178.49 ± 129.83 μm). This study presents the first information on the reproductive cycle of *L. boscii* in the eastern Adriatic Sea and contributes to the understanding of the biology of this species necessary for its sustainable management.

## 1. Introduction

The spawning period is of major importance in fisheries biology, ecology, and management because of its importance for recruitment, survival and stock biomass. The knowledge of fecundity is useful in investigating the population dynamics and can be used for quantification of the fish reproductive capacity [1]. Four-spotted megrim *Lepidorhombus boscii* (Risso, 1810) is a demersal fish distributed on soft bottoms in the northeast Atlantic, and Mediterranean and Black Sea at depths between 60 and 500 m, mostly from 200 to 300 m [2,3,4]. It feeds mainly on pelagic fish, cephalopods and crustaceans [5]. This species is caught exclusively with bottom trawls [6]. In Croatia, *L. boscii* is classified in the category of “megrims” together with other related flatfish such as *L. whiffiagonis* with annual catches of about 5 tons [6]. Like most other commercially important demersal species, *L. boscii* is threatened primarily by intensive bottom trawling and degradation and/or loss of habitat due to fishing [6].

Previous studies on this species were mainly focused on age, growth and mortality [7,8,9,10,11]. In addition, there are several studies on food and feeding habits [5,12,13]. On the other hand, available data on the reproductive biology of this species are very limited. Studies on the reproduction of *L. boscii* providing information on its spawning season, gonadosomatic index (GSI) and fecundity along the Portuguese coasts were conducted by Santos [7,8], Vassilopoulou and Haralabous, and Teixeira et al. [12,13]. Vassilopoulou and Haralabous [12,13], Cengiz et al. [14] and Taylan and Uluturk [15] studied the reproductive biology of this species in the Aegean Sea.

Despite the wide distribution of *L. boscii* in the Adriatic Sea and its commercial importance, knowledge about the biology of this species is still incomplete and very scarce.

The aim of this study was to examine reproductive features, such as sex ratio, size at sexual maturity, spawning period and fecundity of the four-spotted megrim in the eastern Adriatic Sea. Among other things, knowledge about the biology, including reproductive biology, is crucial for stock assessments and long-term sustainable management of this commercially important species, not only in the Adriatic but in the whole area of its distribution.

## 2. Materials and Methods

*L. boscii* specimens (*n* = 963) were collected monthly in the eastern part of the central Adriatic Sea (Figure 1) from July 2020 to June 2021. Samples were collected with a bottom trawl net with 40 mm square mesh cod-end at depths from 90 to 120 m. All samples were transferred to the laboratory for further analysis.

The total body length of the individuals (Lt) was measured with an ichthyometer with an accuracy of 0.1 mm. Body and gonad weight were measured with a digital scale with an accuracy of 0.01 g. The sex and maturity stage of gonads were determined macroscopically based on their appearance, e.g., coloration, size, shape and on relative size of the gonads. Maturity stages were determined according to Holden and Raitt [16] (Table 1).

The obtained data were added to the database in the Microsoft Excel 2010 program. Microsoft Excel 2010 was also used to calculate mean values, standard deviations, and minimum and maximum values. For other statistical analyses, R software was used. Mean length and weight values of females and males were compared using the *t*-test, and the Kolmogorov–Smirnov test determined the representation of total body length and weight for both sexes, while the sex ratio was analyzed using the Chi-square test. Sexual maturity was analyzed in the sizeMath package in R (version R 4.2.2).

The spawning period was determined based on the frequency of occurrence of sexually mature individuals during each month and the analysis of the gonadosomatic index. The gonadosomatic index was calculated using the equation: GSI = (Wg/W) × 100, where GSI is gonadosomatic index, Wg is gonad weight and W is body weight.

The relationship between the proportion of sexually mature specimens and total body length was described by a logistic model according to the equation: Y = 1/(1 + exp(a − b × X)), where Y is the proportion of mature individuals in relation to the total body length, X is total body length (Lt), a and b are parameters of the logistic equation. Based on the parameters a and b of the logistic equation, the lengths at which 25% (LT_25%_), 50% (LT_50%_) and 75% (LT_75%_) of the individuals are sexually mature were determined: LT_25%_ = (a − ln3)/b, LT_50%_ = a/b, LT_75%_ = (a + ln3)/b [17].

Fecundity was determined by dividing the ovarian sample of mature females into three equal parts (anterior, middle, and posterior part). Each subsample was weighed to 0.01 g, and then placed in a petri dish containing a drop of distilled water, the number of oocytes was counted with the stereoscope, and oocyte size was measured in the program AxioVision 4.8. Fecundity was determined based on the following equation by Laevast [18]: F = (Wov/Wu) × Nu, where F is fecundity, Wov is ovary weight, Wu is weight of the selected ovary sample and Nu is the number of oocytes in the selected ovary sample. The relationship between fecundity and total body length, body weight and gonadosomatic index was described by linear models, and the relationship between fecundity and gonad weight was described by a non-linear model.

A subsample of gonads (*n* = 150) was chosen for histological slides. The gonads were previously fixed in a 4% formaldehyde solution, with the aim of preserving morphological and molecular features. Fragments were removed from previously fixed gonads and dehydrated with a series of alcohol solutions that varied from 70% aqueous alcohol to 100% ethanol. After being embedded in ascending solutions of resin, the tissue was sliced longitudinally or cross sections were made with a microtome into 8–10 mm sections, and stained with hematoxylin and eosin. Histological preparations were imaged using a Leica DMC 4500 microscope in the LAS program (Leica Application Suite, Leica Microsystems, Wetzlar and Mannheim, Germany). The stage of gonadal development for each ovary was defined according to the most advanced group of oocytes present in the sample: oocytes in the chromatin nucleolus stage, the perinuclear stage, the cortical alveoli stage, the vitellogenic oocyte, migratory nucleus or hydrated. Oocyte diameter along the longest axis was measured using the program AxioVision 4.8. Statistical differences in oocyte diameter between different developing stages of gonads by season (spring: March, April, May; summer: June, July, August; autumn: September, October, November; and winter: December, January and February) were tested using the Kruskal–Wallis and Dunn tests.

## 3. Results

### 3.1. Size Structure and Sex Ratio

The total sample of *L. boscii* from the Adriatic Sea (*n* = 963) included 508 females (52.7%), and 455 males (47.2%). The total body length and total weight of females and males for each month are shown in Table 1. The mean values of total body lengths of males and females were statistically different (*t*-test, *p* < 0.0005). Differences in total body lengths in both sexes were significantly different (Kolmogorov–Smirnov test; *p* < 0.001) (Figure 2). Differences in weight for both sexes show statistically significant differences (Kolmogorov–Smirnov test; *p* < 0.001). The mean weight of females was significantly higher than the mean weight of males *(t*-test, *p* < 0.0005). Sex ratio analysis showed significant domination of females in February, May, July and August (*p* < 0.05), while for the rest of the months, the m/f ratio was close to 1:1 (Table 2). More males were caught in January, March, April, and August (*p* < 0.05). The sex ratio increased in individuals with the largest total body length. Larger individuals were females. The largest caught female had a total body length of 34 cm and the largest male was 26.1 cm.

### 3.2. Maturation

The length at first sexual maturity was determined on a sample of 963 individuals. The sample consisted of 508 females with a total body length range from 12.50 to 34.00 cm, and 455 males ranging from 13.60 to 26.10 cm. The length at first sexual maturity (*L*_50_) was 19.2 cm (Cl_95%_ = 18.9–19.5) and 20.3 cm (Cl_95%_ = 19.9–20.7) for females and males, respectively (Figure 3).

The relationship between the proportion of mature females (Y) and the total body length (Lt) of *L. boscii* is described by the following logistic equation:Y = 1/(1 + exp (17.16 − 0.89 Lt))

Based on the logistic equation, the lengths at which 25% and 75% of females are sexually mature were determined: Lt_25%_ = 18.04 cm and Lt_75%_ = 20.51 cm.

The relationship between the proportion of mature males (Y) and the total body length (Lt) of *L. boscii* is described by the following logistic equation:Y = 1/(1 + exp (12.08 − 0.59 Lt))

Based on the logistic equation, the lengths at which 25% and 75% of males are sexually mature were determined: Lt_25%_ = 18.61 cm and Lt_75%_ = 22.33 cm.

### 3.3. Gonadogenesis and Spawning Season

The values of the gonadosomatic index for the total sample (*n* = 963) were in the range from 0.02 to 7.46 (0.58 ± 0.90), from 0.02 to 7.46 (0.80 ± 1.13) and from 0.03 to 2.47 (0.33 ± 0.37) for females and males, respectively (Table 3). The mean value of the gonadosomatic index for females (0.80) was statistically significantly higher than the mean value for males (0.37) (*t*-test, *p* < 0.0005). The highest mean values for females were recorded in November, January and March and for males in November and December (Figure 4).

### 3.4. Fecundity

The fecundity of *L. boscii* was analyzed on a sample of 30 mature females with a total body length range from 23.5 to 26.50 cm (25.12 ± 1.00 cm) and a body weight from 99.99 to 179.00 g (125.89 ± 23.34 g). The weight of the gonads ranged from 2.00 to 9.16 g (4.92 ± 2.31 g) and the gonadosomatic index (GSI) ranged from 1.82 to 7.06 (3.83 ± 1.50). The number of mature oocytes ranged from 22.248 to 142.848 (91,910.2 ± 44,285.62) (Table 4).

Relationships of fecundity (F) with total body length (Lt), body weight (W), female gonad weight (Wg) and gonadosomatic index of *L. boscii* are described by the following equations (Figure 5):F = 5.3 × 10^4^ Lt − 10^6^; R^2^ = 0.61(1)
F = 1641.9 W − 9.8 × 10^4^; R^2^ = 0.30(2)
F = 1.9 × 10^3^ Wg ^0.997^; R^2^ = 0.71(3)
F = 3.6 × 10^5^ GSI − 3.0 ∗ 10^4^; R^2^ = 0.62(4)

The results indicated an increase in fecundity with an increase in total body length, body weight, gonad weight and gonadosomatic index of the analyzed individuals. The highest coefficients of determination were recorded for the relationship between fecundity and gonad weight (R^2^ = 0.71) and for the relationship between fecundity and gonadosomatic index (R^2^ = 0.62). A weak positive correlation was determined for the relationship between fecundity and weight (R^2^ = 0.30). Relationships of fecundity with total body length, body weight and gonadosomatic index were described by linear models, while the relationship between fecundity and gonad weight was described by a non-linear power model.

### 3.5. Histology

#### 3.5.1. Gonadal Development in Females

Histological analysis of the ovaries of *L. boscii* revealed the presence of oocytes in the chromatin nucleolus stage, the perinuclear stage, the cotrical alveoli stage and the vitellogenic oocyte stage throughout the year.

During July (Figure 6A), the ovaries contained vitellogenic oocytes (maturation phase) and a large number of mature oocytes filled with yolk droplets, but also oocytes in the stage of nuclear migration. The activity of vitellogenic oocytes was consistent with the values of the gonadosomatic index. In the stage of nuclear migration, the nucleus migrated towards the peripheral part, and the yolk and oil droplets merged. In July, enlarged oocytes are also observed, which rapidly increase their volume, and the cytoplasm remains surrounded by the zona radiata.

During August, oocytes in different stages of development predominated (previtellogenic, vitellogenic and oocytes in the stage of cortical alveoli) (Figure 6B). In the ovaries, the oocytes are in the chromatin nucleus stage, which is characterized by small round cells with a thin and basophilic cytoplasm. Vitellogenic oocytes were represented to a lesser extent.

In September and October, there is an increase in the number of oocytes in the chromatin nucleolus stage, the perinuclear stage, and the cortical alveoli stage (Figure 6C,D), while vitellogenic oocytes are not visible. After this stage, intense growth and development of the gonads occur.

In November, the gonads develop and the stage of vitellogenesis occurs, the appearance of vitellogenic oocytes that are full of yolk and oil droplets (Figure 6E). In addition, mature oocytes in the migration stage are also visible, there is a sudden increase in cell volume due to hydration and limited protein intake. Oocyte activity coincides with gonadosomatic index values.

In December, oocytes were present in the chromatin nucleus stage, the cortical alveoli stage, the vitellogenesis stage, and the stage of nuclear migration (Figure 6F).

In January, after spawning, an increased number of post-vitellogenic oocytes that were not expelled during spawning was visible, so their degeneration and resorption-atresia occurred (Figure 6G). Irregularly shaped germinal tissue and postovulatory bubbles are visible. Furthermore, in the stage of spawning, both previtellogenic and vitellogenic oocytes are visible.

During February, the ovaries were again in the previtalogenesis phase, and the number of oocytes in the chromatin nucleolus stage and the perinuclear stage increased (Figure 6H). After this stage, the more intense development of the gonads begins, as well as the appearance of the vitellogenesis stage.

In March, the intensive development of the gonads begins. The image of the histological preparation shows vitellogenic oocytes full of yolk and oil droplets. In this stage, mature oocytes in the stage of nuclear migration and hydrated oocytes (hydration stage) are observed (Figure 6I). In the hydration stage, water and proteins enter the cell. The diameter of the oocyte increases until the rupture of the nuclear membrane. Postovulatory tissue is also visible in the picture.

In April, oocytes in the chromatin nucleus, perinuclear, cortical alveoli stage and oocytes in the vitellogenesis stage predominate, which are represented in the smallest percentage (Figure 6J).

On the histological preparation from May, atretic oocytes are visible, which are characterized by the disintegration of the nucleus, vitelline globules and zona radiata (Figure 6K).

In June, as well as in April, oocytes in the chromatin nucleus, perinuclear, cortical alveoli stage and oocytes in the vitellogenesis stage predominate (Figure 6L).

Representation of different stages of oocyte development during the one-year cycle is in accordance with the changes recorded in the gonadosomatic index and the analysis of histological preparations. The range of oocyte diameter by month is shown in Table 5. Differences in oocyte diameters between seasons were statistically different (Kruskal–Wallis Chi-squared = 124.12, df = 3, *p* = 0). A statistically significant difference was recorded between all seasons; between spring and autumn, spring and summer, autumn and winter; winter and summer, *p* < 0.00005, and between autumn and summer (*p* = 0.02, *p* < 0.05).

#### 3.5.2. Gonadal Development in Males

The testes of male *L. boscii* are divided into lobes containing cells in different stages of development (spermatogonia, spermatocytes and spermatids). Reproductive cells in all stages of spermatogenesis are visible on histological preparations of the testes throughout the year. Spermatogonia are located next to the basement membrane, while spermatocytes are located closer to the lumen of the seminiferous tubule. Spermatocytes are visible along the edge of the lumen of the seminiferous tubule. Spermatozoids are visible inside the spermatogenic cysts (Figure 7A). During July and August and in the period from November to April, an increase in spermatozoa is visible in the testicles (Figure 7B), but spermatogonia are also visible, which represent reserve material for the next spawning season. Within the spermatogenic cysts, spermatocytes, spermatids and spermatogonia are visible (Figure 7C). From January to March, spermatogonia and spermatocytes are present, as well as the remains of decomposing spermatozoa. In the period from March to June, spermatozoa are visible (Figure 7D).

## 4. Discussion

This study presents the first data on the reproductive characteristics of *L. boscii* in the Adriatic Sea. The sex ratio showed an overall dominance of females in the total sample. The significant domination of females occurred in February, May, July and August while males were dominant in January, March, April and August (*p* < 0.05). When it comes to the sex ratio for the total sample, similar results were obtained by Santos [7] (1:0.95), Castihlo et al. [10] (1:0.76), Teixeira et al. [13] (1:0.93) on the coast of Portugal, Taylan and Uluturk [15] (1:0.41) in the Aegean Sea, and Vassilopoulou et al. [19] (1:0.73) in Greek waters. These differences in the sex ratio could be due to differences in the number of specimens examined, as well as to the area and seasonal effects [15].

Females of *L. boscii* reached their first sexual maturity at smaller total body lengths than males. The length at which 50% of specimens were sexually mature was 19.18 cm for females and 20.25 cm for males. Teixeira et al. [13] also stated that 50% of individuals of *L. boscii* from the Portuguese coast become sexually mature in the second year, i.e., with a total body length of 18.20 to 19.00 cm, while Santos [8], also for the Portuguese coast, stated that females reached their first sexual maturity at 20 cm, i.e., at the age of 3.2 years, and males at 19 cm, i.e., at the age of 2.8 years [8]. In the western Aegean Sea, individuals reach sexual maturity (L50%) with a total body length of 20 cm [20]. Minor differences in L50% in different research areas can be attributed to prevailing biotic and abiotic factors in these regions [13], such as sea temperature, salinity, photoperiod or amount of available food. For instance, many organisms grow faster and mature earlier at higher temperatures than at lower ones (“temperature-size rule”) [21]. Higher temperatures increase the maximal rate of gonadal growth more than they speed the mass-specific rate of assimilation, so the size at maturity should decrease with increasing temperature [22]. However, in all locations (western, central and eastern Mediterranean), small differences in the length of first sexual maturity were recorded. In addition, a decrease in the length of the first sexual maturity can indicate the vulnerability of the species to excessive exploitation which can have negative consequences on the recruitment and preservation of the entire population and thus cause biological responses such as maturation of fish at smaller sizes [23].

The analysis of the gonadosomatic index (GSI) provides a quantitative assessment of the degree of gonad development and spawning season [24]. Based on the gonadosomatic index values, it can be concluded that the species has a prolonged spawning season in the Adriatic Sea with the main spawning season between November and March. The consequence of this spawning pattern is the existence of several “microcohorts” of recruits throughout the year [25]. The peak of spawning is registered in March which corresponds to the results obtained for the species in the South Adriatic Sea [26]. Similar results for the species *L. whiffiagonis* were recorded by Robson [27] for the area of the west coast of Ireland, where the peak of spawning was recorded in March, while the values decrease sharply in August.

The estimated fecundity of *L. boscii* in the Adriatic Sea ranged from 22,248 to 142,848 (87,461 ± 40,677). Similar values (29,977 to 282,937 (89,132 ± 58,703)) were reported by Taylan and Uluturk [15] for the Aegean Sea, and by Santos [7] in the Portuguese coast (42,000–180,000 (119,390 ± 54,912). Fecundity can be influenced by the quantity and quality of food consumed by fish, but also by environmental factors [15]. Egg production in females can vary depending on the species and size of the individual [24]. In this sample, fecundity increases with total body length and higher temperatures. The same was recorded on the Portuguese coast and the Aegean Sea [7,15]. Changes in egg number based on fish length could affect recruitment in fish stocks, especially in exploited fisheries areas [28].

The use of macroscopic and histological assessments to determine the spawning time of marine teleost fish has been widely used [29]. The reproductive status of fish is often determined by the anatomy of the gonads due to the existence of many macroscopic maturity scales [29]. Macroscopic assessment of gonads is a rapid and inexpensive method of determining maturity status and allows for a large number of specimens to be processed on board research vessels and is especially useful where facilities to carry out extensive histological examinations are absent [27]. Although macroscopic staging can allow detailed recording of the seasonal occurrence of different reproductive stages, histological analyzes of gonads are much more precise. Cellular substructures can be identified in developing follicles and ovarian tissue, allowing for a more accurate interpretation of the reproductive status.

Based on histological analysis, it was determined that this species has mature gonads in all months and thus probably spawns through most of the year. Spawning time in the Aegean Sea was prolonged from the winter to spring (February–April) [15]. The study by Vassilopoulou and Haralabous [12] in the north Aegean Sea determined the spawning period as spring. According to Santos, the spawning period of four-spotted megrim occurred between January and March [7]. The differences in spawning time can be attributed to the different temperatures of the seawater from studies of different regions. Throughout the year, the ovaries of *L. boscii* contain oocytes in the stage of previtellogenesis and vitellogenesis. In March and November, the number of oocytes increases, and oocytes in the stage of nuclear migration and hydration are visible. Atresia is visible in January, and postovulatory tissue, but also vitellogenic oocytes in March. This is proof that the species spawns several times during the year. Postovulatory tissue in the gonads is most visible in the period after spawning, that is, in the period after the highest values of the gonadosomatic index have been recorded. In teleost fish, it has been proven that numerous factors such as sea temperature, scarcity of food, presence of stress, etc., cause the appearance of atresia [30]. Luckenbach et al. [31] describe atresia as a regulatory process for maintaining ovarian homeostasis, which ultimately represents the relationship between the number and size of oocytes. Unused oocytes are resorbed after the end of the reproductive cycle [31]. Male reproductive development coincides with female reproductive development. In July and August and from November to April, an increase in spermatozoa is visible in the tubules of the testicles, the number of which continues to increase. In January, spermatogonia and spermatocytes are visible, as well as the remains of spermatozoa with reserve material for the next spawning season. By examining the histological preparations of females and males, it was determined that the development of the gonads is in line with the values of the gonadosomatic index.

The oocyte diameter of *L. boscii* mature females in the Adriatic Sea ranged from 34.01 to 562.72 μm (178.49 ± 129.83 μm). In the ovaries of *L. boscii*, oocytes gradually increase in size along with ovarian development. Previtellogenic oocytes were characterized by large cytoplasm. After the primary growth phase and the beginning of vitellogenesis, characterized by the formation of zone radiate and the appearance of cortical alveoli in the cytoplasm, the oocyte diameter starts to increase more rapidly. In March, oocyte diameters reach a diameter of 562.72 μm. During this period, histological observation revealed that ovaries contain a high presence of oocytes in migrating nucleus, yolk and hydrated stage and in testes a high abundance of spermatozoa. There are no available data on oocytes development and their sizes for *L. boscii*. However, there are data for the closely related species *L. whiffiagonis* from the west coast of Ireland in which the diameter of oocytes of all stages of development was in the range between 27 and 616 μm (209.48 ± 44.74 μm) [27]. In both studies, the range of oocyte diameters was not identified for each individual maturity stage, but it was observed that the oocyte diameters of *L. boscii* from the Adriatic Sea and *L. whiffiagonis* from the west coast of Ireland increased toward spawning time. For both species, a continuous representation of all sizes of oocytes is visible throughout the year. Previtellogenic and vitellogenic oocytes are almost always represented in the gonads. Based on the histological analysis of female gonads and the size distribution of oocytes, it was determined that *L. boscii* has asynchronous ovarian development and spawns several times a year. [32]. Fecundity can be of a specific or unspecific type. In the specific type, no new oocytes are formed during spawning. Species with indeterminate fecundity continuously produce new oocytes for the entire spawning period and it depends on environmental factors (temperature, food availability) and on total body length and weight [32]. It is evident from all that *L. boscii* in the eastern Adriatic has an indeterminate fecundity.

## 5. Conclusions

Mature individuals of *L. boscii* in the Adriatic Sea are present throughout the year. Based on the results of the gonadosomatic index and the histological analysis of the gonads, it can be concluded that this species is a multiple spawner with the most intensive spawning period from November to March. Spawning through most of the year increases the resilience of the population and reduces the risk of overexploitation of the species. The results obtained in this study represent important input data for stock assessment and sustainable management of this commercially important fish species. Furthermore, the results should contribute primarily due to the deficiency of recent relevant studies.

## Figures and Tables

**Figure 1 biology-12-00131-f001:**
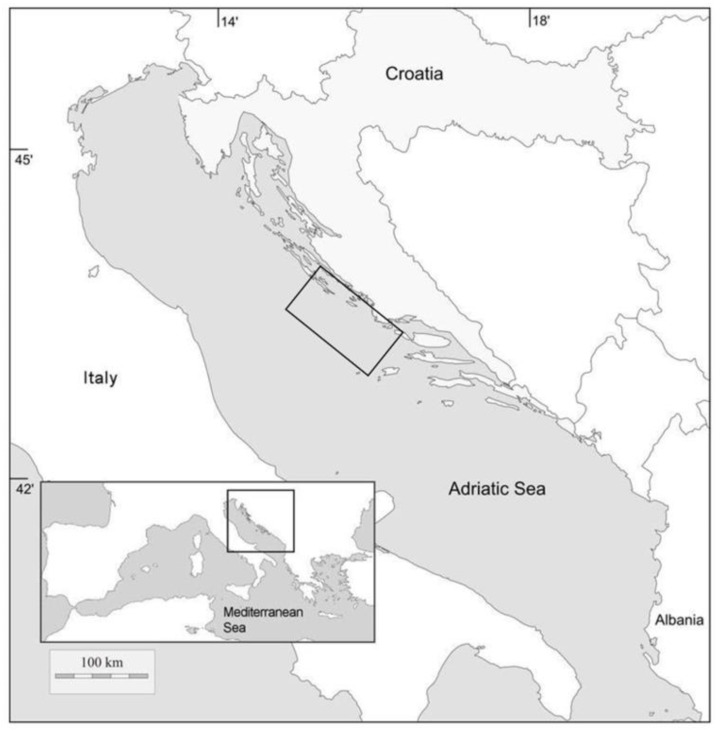
Sampling area of *Lepidorhombus boscii* in the eastern Adriatic Sea.

**Figure 2 biology-12-00131-f002:**
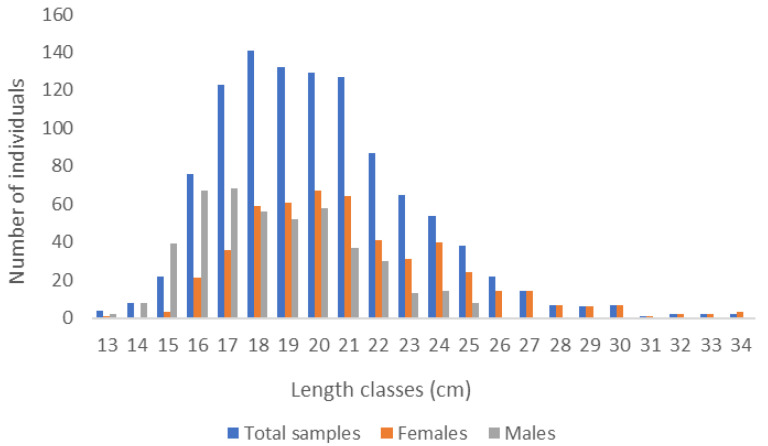
Length distribution of *L. boscii* in the eastern Adriatic Sea (*n* = 963).

**Figure 3 biology-12-00131-f003:**
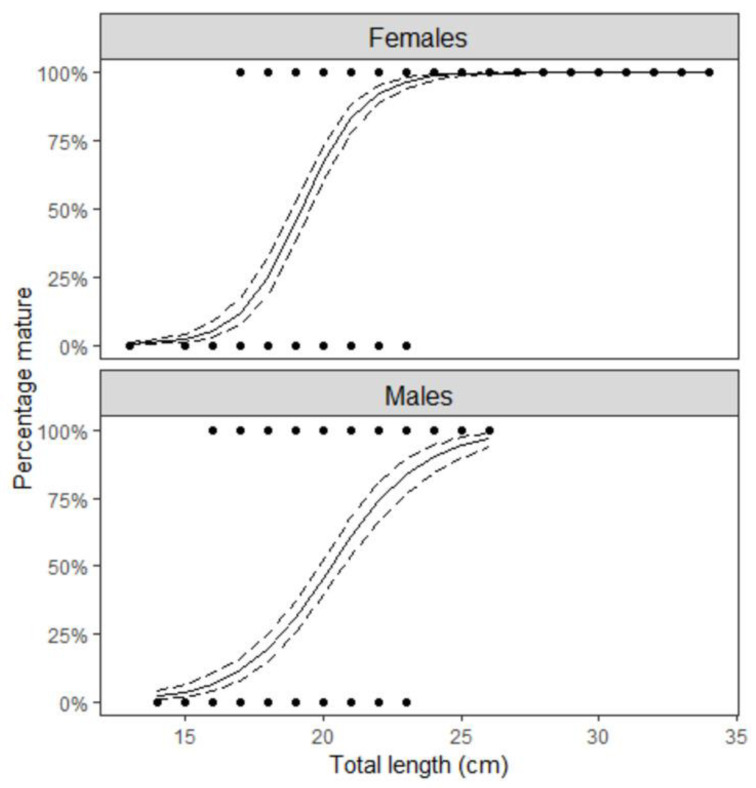
Length at first sexual maturity of *L. boscii* females and males in the eastern Adriatic Sea.

**Figure 4 biology-12-00131-f004:**
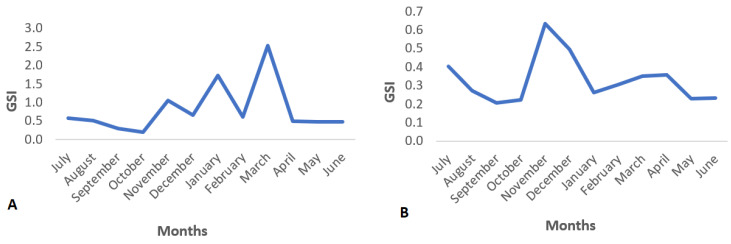
Monthly mean values of gonadosomatic index (GSI%) *in L. boscii* females (**A**) and males (**B**).

**Figure 5 biology-12-00131-f005:**
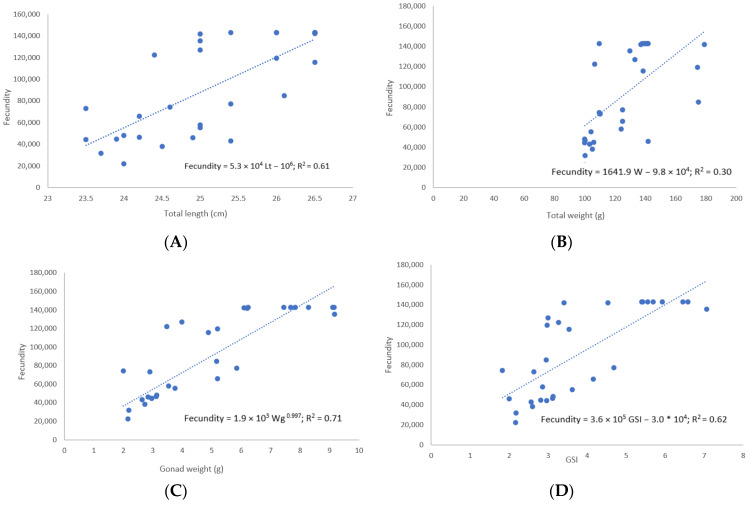
Relationship between fecundity and total body length (**A**), relationship between fecundity and total body weight (**B**), relationship between fecundity and gonad weight (**C**), and relationship between fecundity and gonadosomatic index (**D**).

**Figure 6 biology-12-00131-f006:**
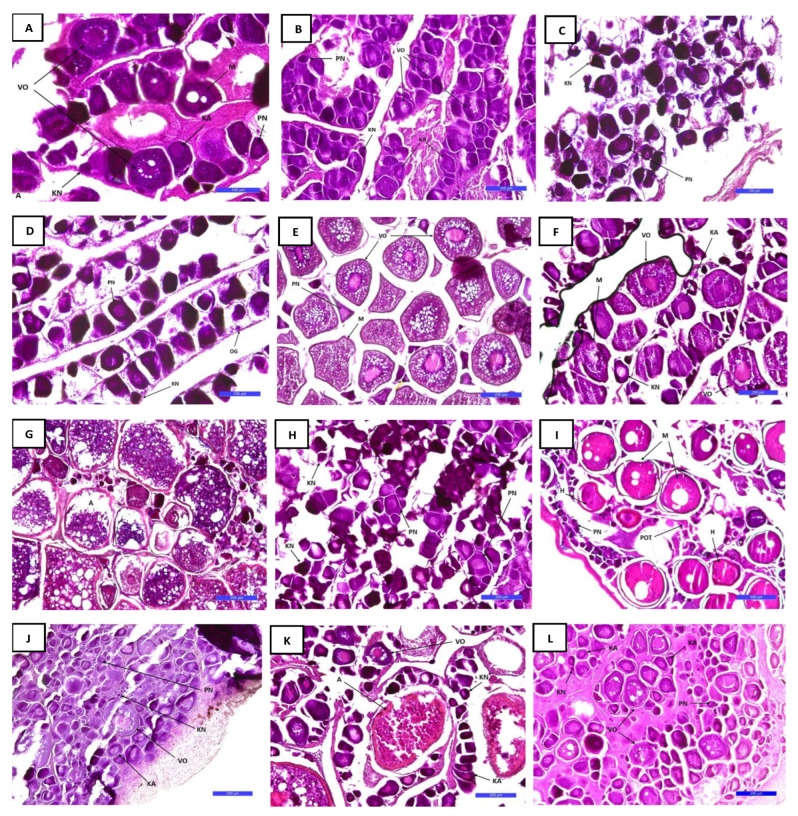
Photomicrographs of the female gonad developmental stages in *Lepidorhombus boscii* by month (from July 2020 (**A**) to June 2021 (**L**)): chromatin nucleolus stage (KN); perinuclear stage (PN); stage of cortical alveoli (KA); stage of vitellogenesis (VO); nuclear migration (M); atresia (A); hydration stage (H); and postovulatory tissue (POT). Scale bar: 100 μm (**A**,**C**,**D**), 200 μm (**B**,**E**–**G**,**H**,**J**–**L**), and 500 μm (**I**).

**Figure 7 biology-12-00131-f007:**
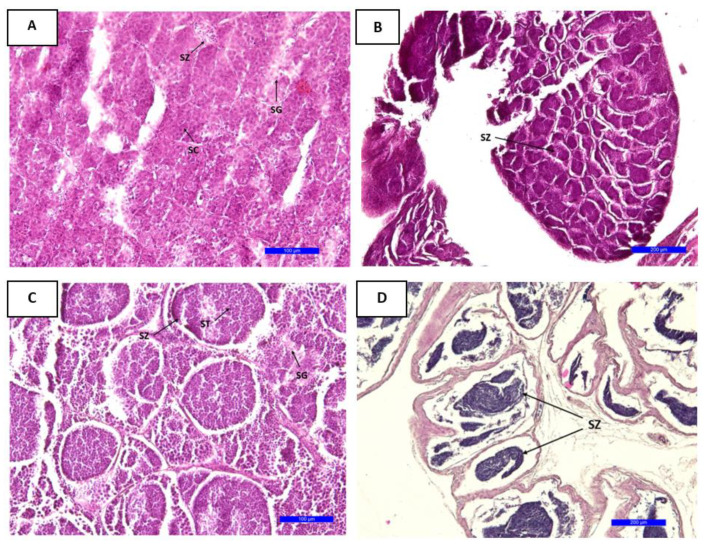
Histological section of testes in *Lepidorhombus boscii* during August (**A**,**B**), November (**C**) and March (**D**): spermatogonia (SG); spermatocytes (SC); spermatids (ST); spermatozoa (SZ). Scale bar: 100 μm (**A**,**C**) and 200 μm (**B**,**D**).

**Table 1 biology-12-00131-t001:** Macroscopically determined maturity stages for ovaries and testes adapted from Holden and Raitt [16].

Stage	State	Description
I	Immature	Ovary and testis about 1/3rd length of body cavity. Ovaries pinkish, translucent; testis whitish. Eggs not visible to naked eye.
II	Maturing virgin and recovering spent	Ovary and testis about 1/2 length of body cavity. Ovary pinkish, translucent; testis whitish, symmetrical. Eggs not visible to naked eye.
III	Ripening	Ovary and testis are about 2/3rds length of body cavity. Ovary pinkish-yellow color with granular appearance, testis whitish to creamy. No transparent or translucent eggs visible.
IV	Ripe	Ovary and testis from 2/3rds to full length of body cavity. Ovary orange-pink in color with conspicuous superficial blood vessels. Large transparent, ripe eggs visible. Testis whitish-creamy, soft.
V	Spent	Ovary and testis shrunken to about 1/2 length of body cavity. Walls loose. Ovary may contain remnants of disintegrating opaque and ripe eggs, darkened or translucent. Testis bloodshot and flabby.

**Table 2 biology-12-00131-t002:** Number of specimens, sex ratio, total length (Lt) range of males and females *L. boscii* for each month (*—statistical significance, *p* < 0.05).

Months	N			Sex Ratio	Lt Range (cm)		W Range (g)	
	Total	Males	Females		Males	Females	Males	Females
July	99	36	63	1:1.75 *	16.0–25.0	14.5–34.0	22.6–126.2	20.4–288.2
August	86	44	39	1:1.12 *	15.9–22.0	16.9–30.1	26.6–75.5	29.1–210.5
September	50	15	26	1:0.57	16.9–21.9	16.1–21.3	29.4–69.2	24.8–60.3
October	86	37	49	1:0.75	13.6–21.9	16.6–20.5	14.4–74.1	27.9–59.6
November	94	41	44	1:0.93	17.0–26.1	16.7–27.5	30.7–117.0	27.5–177.3
December	76	37	37	1:1	15.8–23.4	16.2–21.6	23.5–79.3	25.6–62.0
January	104	43	39	1:1.10 *	16.0–25.7	17.9–26.5	26.3–113.8	34.6–175.1
February	80	27	45	1:0.6 *	16.3–22.9	16.0–23.5	27.5–85.2	25.0–103.8
March	108	45	40	1:1.12 *	14.9–25.5	18.4–26.6	22.3–112.9	38.6–139.2
April	99	56	28	0.5:1 *	14.6–21.0	14.5–22.0	21.9–63.8	21.4–77.5
May	117	43	61	1:0.70 *	15.5–26.0	15.8–34.0	26.6–129.7	26.4–280.0
June	70	31	37	1:0.83	14.2–25.7	12.5–24.8	20.4–126.3	12.1–120.1

**Table 3 biology-12-00131-t003:** Range and mean (±SD) gonadosomatic index (GSI) of *L. boscii* females and males.

Females				Males		
Month	N	Range	$\bar{x}$ ± SD	N	Range	$\bar{x}$ ± SD
July	63	0.02–2.92	0.58 ± 0.56	36	0.10–2.00	0.40 ± 0.39
August	38	0.09–1.81	0.51 ± 0.43	44	0.17–1.93	0.27 ± 0.28
September	26	0.09–0.59	0.30 ± 0.14	15	0.08–0.62	0.20 ± 0.14
October	49	0.09–0.45	0.20 ± 0.07	36	0.05–1.31	0.22 ± 0.21
November	44	0.03–3.27	1.05 ± 0.82	41	0.07–1.82	0.63 ± 0.57
December	37	0.09–3.29	0.66 ± 0.60	36	0.11–2.31	0.49 ± 0.44
January	39	0.08–7.06	1.73 ± 2.01	43	0.04–1.05	0.26 ± 0.17
February	45	0.50–5.19	0.60 ± 1.21	27	0.10–2.47	0.30 ± 0.48
March	40	0.09–7.46	2.53 ± 1.88	43	0.09–2.33	0.35 ± 0.47
April	28	0.11–2.05	0.49 ± 0.46	52	0.09–2.19	0.35 ± 0.40
May	61	0.10–1.07	0.48 ± 0.22	42	0.03–0.73	0.23 ± 0.16
June	37	0.18–1.99	0.47 ± 0.31	29	0.05–0.61	0.23 ± 0.16

**Table 4 biology-12-00131-t004:** Mean values of fecundity (F), body weight (W), ovary weight and gonadosomatic index (GSI) by the length class. The total body lengths of the individuals are sorted into 1 cm length classes, e.g., the length class 24 included individuals with total body lengths from 23.5 to 24.4 cm.

Length Class (cm)	W (g)	Ovary Weight (g)	GSI	Fecundity (F)	N
24	105.24 ± 8.25	3.12 ± 0.88	2.93 ± 0.60	55,479 ± 29,355	9
25	121.76 ± 16.57	4.50± 2.54	3.60 ± 1.75	81,695 ± 45	9
26	148.18 ± 27.94	5.76 ± 1.19	4.03 ± 1.26	119,420 ± 27	7
27	139.32 ± 1.29	7.77± 1.75	5.57 ± 1.23	137,359 ± 122,273	5

**Table 5 biology-12-00131-t005:** Oocyte diameter (N = 360) by month in ovaries of mature females of *L. boscii* in the Adriatic Sea.

Months	Range (μm)	$\bar{x}$ ± SD
July	66.0–128.9	99.9 ± 16.2
August	76.8–124.2	102.1 ± 12.9
September	34.0–63.9	53.4 ± 7.1
October	46.5–74.7	61.2 ± 7.6
November	190.2–300.6	243.4 ± 31.6
December	113.8–281.6	203.9 ± 48.2
January	126.2–3687	244 ± 74.9
February	53.2–127.2	86.9 ± 18.2
March	300.6–562.8	419.5 ± 71.9
April	80.0–202.7	117.9 ± 34.6
May	210.7–517.9	401.6 ± 101.1
June	72.9–165.6	107.5 ± 25.9

## Data Availability

Not applicable.

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
