# Peer review of "Fecundity, Length at First Sexual Maturity and Gonadal Development of Lepidorhombus boscii in the Eastern Adriatic Sea"

_biology, 2023, doi:10.3390/biology12010131_

Round 1
Reviewer 1 Report
MS is well-written and planned. The introduction is concise and informative. Materials and methods are lacking some of the vital points- namely, equations used to calculate lengths at maturity are not references and are not ones widely used. Furthermore, that and some other equations are written as they are used in R or excel not mathematically correct. results have the potential to be very informative but are very confusedly written. Possibly additional tables or figures comparing some of the data could be of use. On the other hand, the histological part of the results is rather extensive compared to the other results. therefore that part is additionally confusing and too declarative and very difficult to follow. Discussion is ok although I would suggest taking another look at the literature and maybe adding some of the data from other areas (Iberian Sea, Ionian Sea..).
I think that this paper could add to the knowledge pool but should be revised.

Author Response
Dear Reviewer,
We would like to thank you for the valuable feedback. We studied carefully all the comments and tried to meet the expectations as best as possible in order to improve the manuscript.
Authors
Response to Reviewer 1 Comments
Point 1: MS is well-written and planned. The introduction is concise and informative. Materials and methods are lacking some of the vital points- namely, equations used to calculate lengths at maturity are not references and are not ones widely used. Furthermore, that and some other equations are written as they are used in R or excel not mathematically correct. results have the potential to be very informative but are very confusedly written. Possibly additional tables or figures comparing some of the data could be of use. On the other hand, the histological part of the results is rather extensive compared to the other results. therefore that part is additionally confusing and too declarative and very difficult to follow. Discussion is ok although I would suggest taking another look at the literature and maybe adding some of the data from other areas (Iberian Sea, Ionian Sea..). I think that this paper could add to the knowledge pool but should be revised.
Response 1: We fixed a mathematical error in the formula for sexual maturity. We have improved the structure of the paragraph related to histology, and the discussion, however, we included all available literature on the reproductive biology of Lepidorhombus sp.

Reviewer 2 Report
The aim of the manuscript entitled "Reproductive cycle, size at maturity and fecundity of Lepidorhombus boscii in the eastern Adriatic" was to investigate the reproductive cycle of the four-spotted megrim from the central eastern Adriatic. The topic is particularly relevant to this field because it brings novelty about the reproductive biology of this commercially important species. The manuscript is generally presented in a well-structured manner, easy to read and understand. The data obtained and the analysis of the results carried out are presented in an appropriate manner that will surely improve the current knowledge about the reproductive properties of Lepidorhombus boscii.
The introduction, especially the methods and results sections, are well organized and written in detail. All used experimental methods and statistical analyses are very well explained and sufficient for the interpretation of the results. The presented tables and figures are clear and easy to follow and understand. While the photomicrographs showing the development of the female gonads (Figure 6) are much better stained, more precise in terms of the histological representation of the structures of the gonads and most importantly correctly marked, on the photomicrographs of the development of the male gonads (Figure 7) it is not possible to clearly see the position and morphological difference between spermatocytes (SC), spermatids (ST) and spermatozoa (SZ) (Fig. 7 A, C). It is recommended to repeat histological staining with hematoxylin and eosin in connection with the development of the male gonads in order to achieve a better visualization of the cells at different stages of development.
The discussion should be improved primarily not by repeating the results, which are very well elaborated and interpreted in the Results section, but rather by using evidence and arguments for critical comparison with similar conducted studies with appropriate explanations.
Line 277-278: more detailed explanation needed regarding investigated species
Line 289: elaborate the influence of the specific biotic and abiotic factors
Line 302 – 306: elaborate why you came up with stated conclusion
Line 330 – 333: elaborate why you came up with stated conclusion
Line 361 – 364: elaborate the difference between the oocyte diameter of L. boscii and L. whiffiagonis
The cited literature is not recent, mostly due to the lack of recent relevant studies and consequently the absence of new evidence in the researched field of manuscripts, but it is relevant for the answer to the main research question. The drawn conclusion could be further supported by the obtained results and would sound much better if the main findings of the research were emphasized more.
Author Response
Dear Reviewer
We would like to thank you for the valuable feedback. We studied carefully all the comments and tried to meet the expectations as best as possible in order to improve the manuscript.
Authors
Responses:
Response to Reviewer 2 Comments
Point 1: The aim of the manuscript entitled "Reproductive cycle, size at maturity and fecundity of Lepidorhombus boscii in the eastern Adriatic" was to investigate the reproductive cycle of the four-spotted megrim from the central eastern Adriatic. The topic is particularly relevant to this field because it brings novelty about the reproductive biology of this commercially important species. The manuscript is generally presented in a well-structured manner, easy to read and understand. The data obtained and the analysis of the results carried out are presented in an appropriate manner that will surely improve the current knowledge about the reproductive properties of Lepidorhombus boscii. The introduction, especially the methods and results sections, are well organized and written in detail. All used experimental methods and statistical analyses are very well explained and sufficient for the interpretation of the results. The presented tables and figures are clear and easy to follow and understand. While the photomicrographs showing the development of the female gonads (Figure 6) are much better stained, more precise in terms of the histological representation of the structures of the gonads and most importantly correctly marked, on the photomicrographs of the development of the male gonads (Figure 7) it is not possible to clearly see the position and morphological difference between spermatocytes (SC), spermatids (ST) and spermatozoa (SZ) (Fig. 7 A, C). It is recommended to repeat histological staining with hematoxylin and eosin in connection with the development of the male gonads in order to achieve a better visualization of the cells at different stages of development.
Response 1: We repeated the histological staining on the stored samples, while some histological preparations were re-photographed. We have replaced Figure 7A with another male from that month, because the cells are better seen.
Point 2: The discussion should be improved primarily not by repeating the results, which are very well elaborated and interpreted in the Results section, but rather by using evidence and arguments for critical comparison with similar conducted studies with appropriate explanations.
Response 2: We have analyzed your comment and improved the discussion in the best possible way.
Other responses:
Line 277-278: more detailed explanation needed regarding investigated species - corrected
Line 289: elaborate the influence of the specific biotic and abiotic factors - Corrected! We have elaborated on the main abiotic factors, i.e., differences in temperature and salinity between the Adriatic and Aegean Sea.
Line 302 – 306: elaborate why you came up with stated conclusion - We quoted these claims from the mentioned studies, and the authors obtained these values from the monthly gonadosomatic index.
Line 330 – 333: elaborate why you came up with stated conclusion - The analysis of histological preparations confirmed the peak of spawning for both sexes. The cited authors did not perform a histological analysis of the gonads, just a GSI analysis, but the highest values were recorded in the same season.
Line 361 – 364: elaborate the difference between the oocyte diameter of L. boscii and L. whiffiagonis - We have explained the differences in lines 418-430.
The cited literature is not recent, mostly due to the lack of recent relevant studies and consequently the absence of new evidence in the researched field of manuscripts, but it is relevant for the answer to the main research question. The drawn conclusion could be further supported by the obtained results and would sound much better if the main findings of the research were emphasized more. - Thanks for this advice. We improved the conclusion.

Reviewer 3 Report
Page 1, line 18: all the authors confident that they can measure the diameter to 0.01µm?
Page 1 line 36: “there” typing.
Page 2, line 53: Perhaps some reference here the total number of fish would help.
Page 2, line 54: please give details of the bottom trawl net in terms of mesh size as this will affect sampling profile?
Page 2 line 55: were the fish measured at sea or in the laboratory
Page 2, line 56: please give some information of how the gonad
Page 2, line 57: is it necessary to measure body weight to the nearest 0.01g? I suggest using fish weight to the nearest 0.1g. As whether the fish is wet and covered in mucus makes use of 0.01 g arbitrary?
Table 1: the legend of the table refers to ovaries whether there are references to testes in various stages please clarify?
Page 3, line 65: the obtained data were analysed using Microsoft Excel 2010
Page 6, line 153: the gonado- somatic index appears to be very variable samples taken.
In November for the males mean value is below the lower range value of 0.7? This appears to be a peculiarity of the males. Please explain this statistically.
Page 6, line 159: Is there an explanation why there is no synchrony between the highest GSI levels in males and females?
Table 4: Could the fecundity be rounded up without a decimal point. Could the authors please explain the large differences in variability in the fecundity values between the different length classes?
Page 7, line 194: Please restructure this paragraph splitting it up into subheadings to make it more readable. As it stands it is a long description of the results without any contextual implications.
Figure 6: okay scale bars are 100 200 and 500 but we need to know for which figure or is this always in a horizontal line for instance a, b and c?
Table 5: Again reduce numbers to 1 decimal place. Again there appears to be large variation between different months for instance in February we have a mean of 86µm and then in March 419µm
Figure 7: Again scale bars?
Page 10, line 272: This is the case for the Adriatic but there is much published data for Portuguese waters.
Page 11, line 320: I would have thought that GSI is a much more competent measure for maturity? Than a subjective macroscopic scale.
Page 12, line 367: that the most intensive spawning time(s)?... March and November?
It would be helpful for the authors to comment on the variability of the data and the rapid changes from month to month in terms of oocyte size. Also some consideration of multi-spawners, batch spawning and what are the advantages of spawning all the year-round?
Author Response
Dear Reviewer,
We would like to thank you for the valuable feedback. We studied carefully all the comments and tried to meet the expectations as best as possible in order to improve the manuscript.
Authors
Response to Reviewer 3 Comments
Page 1, line 18: all the authors confident that they can measure the diameter to 0.01µm? - We measured the oocyte in the AxioVision program, which offers the option of precise measurement in μm. This program has been used in several literatures and we hope it is suitable.
Page 1 line 36: “there” typing. - corrected
Page 2, line 53: Perhaps some reference here the total number of fish would help. – We have added the data N=963, but we have already given that information in the first sentence in the Results (line 125)
Page 2, line 54: please give details of the bottom trawl net in terms of mesh size as this will affect sampling profile? – The samples were collected using bottom trawl net with 40 mm square mesh codend. Added!
Page 2 line 55: were the fish measured at sea or in the laboratory - In the laboratory. We added this information.
Page 2, line 56: please give some information of how the gonad
Page 2, line 57: is it necessary to measure body weight to the nearest 0.01g? I suggest using fish weight to the nearest 0.1g. As whether the fish is wet and covered in mucus makes use of 0.01 g arbitrary? – We know that a precision of 0.1g is sufficient for the weight of the fish, but the digital scale we used gives the value to two decimal places. We hope that's okay!
Table 1: the legend of the table refers to ovaries whether there are references to testes in various stages please clarify? - The table describes the macroscopic appearance of ovaries and testes; we have corrected the name of the table/legend.
Page 3, line 65: the obtained data were analysed using Microsoft Excel 2010 - corrected
Page 6, line 153: the gonado- somatic index appears to be very variable samples taken. In November for the males mean value is below the lower range value of 0.7? This appears to be a peculiarity of the males. Please explain this statistically. - We are sorry, this is a mistake. The lower value of the range is 0.07, so the mean is not less than the lower range.
Page 6, line 159: Is there an explanation why there is no synchrony between the highest GSI levels in males and females? - The explanation is in lines 332-335.
Table 4: Could the fecundity be rounded up without a decimal point. Could the authors please explain the large differences in variability in the fecundity values between the different length classes? - Yes, we removed the decimal points. We have explained the connection between fish length and fecundity in lines 359-368.
Page 7, line 194: Please restructure this paragraph splitting it up into subheadings to make it more readable. As it stands it is a long description of the results without any contextual implications. – corrected; instead of subheadings we divided the section in several paragraphs to be easier to follow.
Figure 6: okay scale bars are 100 200 and 500 but we need to know for which figure or is this always in a horizontal line for instance a, b and c? - correced
Table 5: Again reduce numbers to 1 decimal place. Again there appears to be large variation between different months for instance in February we have a mean of 86µm and then in March 419µm - corrected
Figure 7: Again scale bars? -corrected
Page 10, line 272: This is the case for the Adriatic but there is much published data for Portuguese waters. - That's right, in the Introduction we listed previous research, and in the Discussion we compared it with data from other areas of distribution, including the Portuguese waters.
Page 11, line 320: I would have thought that GSI is a much more competent measure for maturity? Than a subjective macroscopic scale. - We emphasized that the analysis of gonadosomatic index provides an information of the degree of gonad development and spawning season. It is often the case that the gonadosomatic index is calculated after sex determination based on macroscopic analysis, without performing a histological analysis.
Page 12, line 367: that the most intensive spawning time(s)?... March and November? - Considering the GSI and histological analysis, we conclude that L. boscii in the Adriatic Sea has a prolonged spawning season in the Adriatic Sea with the main spawning season between November and March, after which there is a drop in value, which is logical considering the condition of the fish after spawning.
It would be helpful for the authors to comment on the variability of the data and the rapid changes from month to month in terms of oocyte size. Also some consideration of multi-spawners, batch spawning and what are the advantages of spawning all the year-round? - We have added an explanation in lines 441-446

Reviewer 4 Report
Line 13, "%" repeated
Introduction is very brief and does not describe the importance of this species and in this study area.
Line 46. "to investigate" doesn't seem like an adecuate objective for an article.
Línea 126. Indicate significant differences in the table.
Line 128. Authors doesnt
Line 128 the method of first maturity length was not explained, why the range 12.50 to 34.00 cm for females and 13.60 to 26.10 for males was chosen.
Figure 4. You can reduce decimal places on the y-axis.
Author Response
Dear Reviewer,
We would like to thank you for the valuable feedback. We studied carefully all the comments and tried to meet the expectations as best as possible in order to improve the manuscript.
Authors
Response to Reviewer 4 Comments
Line 13, "%" repeated - corrected
Introduction is very brief and does not describe the importance of this species and in this study area. - Corrected. We have improved the Introduction
Line 46. "to investigate" doesn't seem like an adecuate objective for an article. - corrected
Línea 126. Indicate significant differences in the table. - corrected
Line 128. Authors doesnt the method of first maturity length was not explained, why the range 12.50 to 34.00 cm for females and 13.60 to 26.10 for males was chosen. - The range of analyzed individuals (total sample) was from 12.50 to 34.00 cm, the range of females was also in this range, and the range for males was slightly smaller, i.e. it was in the range of 13.60 to 26.10 cm. A sample of 963 individuals was analyzed, the word subsample in the text is a mistake and we apologize for that.
Figure 4. You can reduce decimal places on the y-axis. - corrected
* We reviewed the entire manuscript and improved the English language!

Round 2
Reviewer 1 Report
MS is now better constructed and additional information provided adds to the content and understanding. However, I still have an issue with the histology section- I believe it is too detailed compared with the rest of the MS. Therefore I would suggest either shortening it (lines from 211to 259) or changing the topic /title in a way that development of the spawning products is the main topic.
Also please change lines from 190 to 195 in more readable numbers, for example F=5.3 * 104 Lt - 106; F=3.6 *105 GSI-3.0*104.
Author Response
Dear Reviewer,
We would like to thank you for the valuable feedback. We studied carefully all the comments and tried to meet the expectations as best as possible in order to improve the manuscript.
Authors
MS is now better constructed and additional information provided adds to the content and understanding. However, I still have an issue with the histology section- I believe it is too detailed compared with the rest of the MS. Therefore I would suggest either shortening it (lines from 211to 259) or changing the topic /title in a way that development of the spawning products is the main topic. - We have made changes to the main title so that gonadal development is part of the main topic.
Also please change lines from 190 to 195 in more readable numbers, for example F=5.3 * 104 Lt - 106; F=3.6 *105 GSI-3.0*104. - corrected!

Reviewer 2 Report
I thank the authors for answering to my comments. The manuscript is now improved. The position and morphological difference between spermatocytes (SC), spermatids (ST) and spermatozoa (SZ) (Fig. 7 A, C) are now more visible. My suggestions are to include a “temperature–size rule” in lines 314 – 318 and 360 – 361 which explains why organisms grow faster and mature earlier, at warmer temperatures than at cooler temperatures (Arendt, 2011; Brunel and Dickey-Collas, 2010; Matta et al., 2016) and why adaptive changes in energy allocation to reproduction are happened (Atkinson, 1996; Angilletta et al., 2004; Hosono, 2011).
Author Response
Dear Reviewer,
We would like to thank you for the valuable feedback. We studied carefully all the comments and tried to meet the expectations as best as possible in order to improve the manuscript. We included the temperature-size rule and cited some of the authors you suggested.
Authors

Reviewer 4 Report
Authors have made the suggested changes; however, the references are not recent, the ones added in the new version are from 1998, 1997, and the most recent from 2009, I suggest adding references from at least the last 5 years.
Author Response
Dear Reviewer,
We would like to thank you for the valuable feedback. We studied carefully all the comments and tried to meet the expectations as best as possible in order to improve the manuscript. We tried to include as much recent literature as possible. We did not change Sparre and Venema (1998) because the reference refers to the formula for calculating the length of the first sexual maturity. The above formula follows the sizeMath package (2020.) used in the R program. Then, the study by Ungaro and Martino (1998) refers to L. boscii from the western part of the Adriatic Sea, there is no recent literature on the reproductive biology of this species, especially when it comes to the Adriatic Sea.
Authors
